# Coordinate-wise Power Method

**Qi Lei** [1]     **Kai Zhong** [1]     **Inderjit S. Dhillon** [1,2]

[1] Institute for Computational Engineering & Sciences    [2] Department of Computer Science
University of Texas at Austin
{leiqi, zhongkai}@ices.utexas.edu, inderjit@cs.utexas.edu

## Abstract

In this paper, we propose a coordinate-wise version of the power method from an optimization viewpoint. The vanilla power method simultaneously updates all the coordinates of the iterate, which is essential for its convergence analysis. However, different coordinates converge to the optimal value at different speeds. Our proposed algorithm, which we call coordinate-wise power method, is able to select and update the most *important* $k$ coordinates in $O(kn)$ time at each iteration, where $n$ is the dimension of the matrix and $k \leq n$ is the size of the active set. Inspired by the "greedy" nature of our method, we further propose a greedy coordinate descent algorithm applied on a non-convex objective function specialized for symmetric matrices. We provide convergence analyses for both methods. Experimental results on both synthetic and real data show that our methods achieve up to 23 times speedup over the basic power method. Meanwhile, due to their coordinate-wise nature, our methods are very suitable for the important case when data cannot fit into memory. Finally, we introduce how the coordinate-wise mechanism could be applied to other iterative methods that are used in machine learning.

## 1  Introduction

Computing the dominant eigenvectors of matrices and graphs is one of the most fundamental tasks in various machine learning problems, including low-rank approximation, principal component analysis, spectral clustering, dimensionality reduction and matrix completion. Several algorithms are known for computing the dominant eigenvectors, such as the power method, Lanczos algorithm [14], randomized SVD [2] and multi-scale method [17]. Among them, the power method is the oldest and simplest one, where a matrix $A$ is multiplied by the normalized iterate $\boldsymbol{x}^{(l)}$ at each iteration, namely,

$$\boldsymbol{x}^{(l+1)} = \text{normalize}(A\boldsymbol{x}^{(l)}).$$

The power method is popular in practice due to its simplicity, small memory footprint and robustness, and particularly suitable for computing the dominant eigenvector of large sparse matrices [14]. It has applied to PageRank [7], sparse PCA [19, 9], private PCA [4] and spectral clustering [18]. However, its convergence rate depends on $|\lambda_2|/|\lambda_1|$, the ratio of magnitude of the top two dominant eigenvalues [14]. Note that when $|\lambda_2| \approx |\lambda_1|$, the power method converges slowly.

In this paper, we propose an improved power method, which we call coordinate-wise power method, to accelerate the vanilla power method. Vanilla power method updates all $n$ coordinates of the iterate simultaneously even if some have already converged to the optimal. This motivates us to develop new algorithms where we select and update a set of important coordinates at each iteration. As updating each coordinate costs only $\frac{1}{n}$ of one power iteration, significant running time can be saved when $n$ is very large. We raise two questions for designing such an algorithm.

*The first question: how to select the coordinate?* A natural idea is to select the coordinate that will change the most, namely,

$$\operatorname{argmax}_i |c_i|, \text{ where } \boldsymbol{c} = \frac{A\boldsymbol{x}}{\boldsymbol{x}^T A\boldsymbol{x}} - \boldsymbol{x}, \tag{1}$$

where $\frac{A\boldsymbol{x}}{\boldsymbol{x}^T A\boldsymbol{x}}$ is a scaled version of the next iterate given by power method, and we will explain this special scaling factor in Section 2. Note that $c_i$ denotes the $i$-th element of the vector $\boldsymbol{c}$. Instead of choosing only one coordinate to update, we can also choose $k$ coordinates with the largest $k$ changes in $\{|c_i|\}_{i=1}^n$. We will justify this selection criterion by connecting our method with greedy coordinate descent algorithm for minimizing a non-convex function in Section 3. With this selection rule, we are able to show that our method has global convergence guarantees and faster convergence rate compared to vanilla power method if $k$ satisfies certain conditions.

*Another key question: how to choose these coordinates without too much overhead?* How to efficiently select important elements to update is of great interest in the optimization community. For example, [1] leveraged nearest neighbor search for greedy coordinate selection, while [11] applied partially biased sampling for stochastic gradient descent. To calculate the changes in Eq (1) we need to know all coordinates of the next iterate. This violates our previous intention to calculate a small subset of the new coordinates. We show, by a simple trick, we can use only $O(kn)$ operations to update the most important $k$ coordinates. Experimental results on dense as well as sparse matrices show that our method is up to 8 times faster than vanilla power method.

*Relation to optimization.* Our method reminds us of greedy coordinate descent method. Indeed, we show for symmetric matrices our coordinate-wise power method is similar to greedy coordinate descent for rank-1 matrix approximation, whose variants are widely used in matrix completion [8] and non-negative matrix factorization [6]. Based on this interpretation, we further propose a faster greedy coordinate descent method specialized for symmetric matrices. This method achieves up to 23 times speedup over the basic power method and 3 times speedup over the *Lanczos* method on large real graphs. For this non-convex problem, we also provide convergence guarantees when the initial iterate lies in the neighborhood of the optimal solution.

*Extensions.* With the coordinate-wise nature, our methods are very suitable to deal with the case when data cannot fit into memory. We can choose a $k$ such that $k$ rows of $A$ can fit in memory, and then fully process those $k$ rows of data before loading the RAM (random access memory) with a new partition of the matrix. This strategy helps balance the data processing and data loading time. The experimental results show our method is 8 times faster than vanilla power method for this case.

The paper is organized as follows. Section 2 introduces coordinate-wise power method for computing the dominant eigenvector. Section 3 interprets our strategy from an optimization perspective and proposes a faster algorithm. Section 4 provides theoretical convergence guarantee for both algorithms. Experimental results on synthetic or real data are shown in Section 5. Finally Section 6 presents the extensions of our methods: dealing with out-of-core cases and generalizing the coordinate-wise mechanism to other iterative methods that are useful for the machine learning community.

## 2   Coordinate-wise Power Method

The classical power method (PM) iteratively multiplies the iterate $\boldsymbol{x} \in \mathbb{R}^n$ by the matrix $A \in \mathbb{R}^{n \times n}$, which is inefficient since some coordinates may converge faster than others. To illustrate this

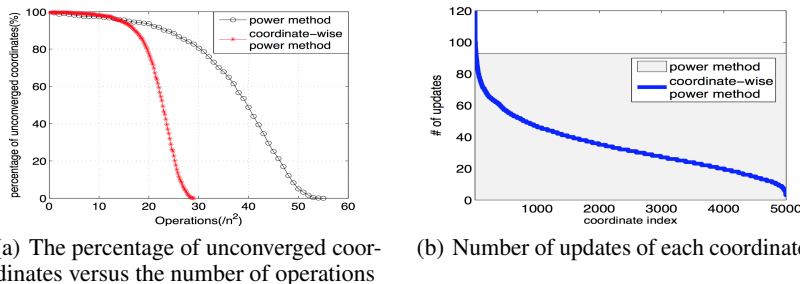

(a) The percentage of unconverged coordinates versus the number of operations

(b) Number of updates of each coordinate

Figure 1: **Motivation for the Coordinate-wise Power Method.**   Figure 1(a) shows how the percentage of unconverged coordinates decreases with the number of operations. The gradual decrease demonstrates the unevenness of each coordinate as the iterate converges to the dominant eigenvector. In Figure 1(b), the X-axis is the coordinate indices of iterate $\boldsymbol{x}$ sorted by their frequency of updates, which is shown on the Y-axis. The area below each curve approximately equals the total number of operations. The given matrix is synthetic with $|\lambda_2|/|\lambda_1| = 0.5$, and terminating accuracy $\epsilon$ is set to be 1e-5.

phenomenon, we conduct an experiment with the power method; we set the stopping criterion as $\|\boldsymbol{x} - \boldsymbol{v}_1\|_\infty < \epsilon$, where $\epsilon$ is the threshold for error, and let $\boldsymbol{v}_i$ denote the $i$-th dominant eigenvector (associated with the eigenvalue of the $i$-th largest magnitude) of A in this paper. During the iterative process, even if some coordinates meet the stopping criterion, they still have to be updated at every iteration until uniform convergence. In Figure 1(a), we count the number of unconverged coordinates, which we define as $\{i : i \in [n]\big||x_i - v_{1,i}| > \epsilon\}$, and see it gradually decreases with the iterations, which implies that the power method makes a large number of unnecessary updates. In this paper, for computing the dominant eigenvector, we exhibit a coordinate selection scheme that has the ability to select and update "important" coordinates with little overhead. We call our method *Coordinate-wise Power Method* (CPM). As shown in Figure 1(a) and 1(b), by selecting important entries to update, the number of unconverged coordinates drops much faster, leading to an overall fewer flops.

---

**Algorithm 1** Coordinate-wise Power Method

---

1: **Input:** Symmetric matrix $A \in \mathbb{R}^{n \times n}$, number of selected coordinates $k$, and number of iterations, $L$.
2: Initialize $\boldsymbol{x}^{(0)} \in \mathbb{R}^n$ and set $\boldsymbol{z}^{(0)} = A\boldsymbol{x}^{(0)}$. Set coordinate selecting criterion $\boldsymbol{c}^{(0)} = \boldsymbol{x}^{(0)} - \frac{\boldsymbol{z}^{(0)}}{(\boldsymbol{x}^{(0)})^T \boldsymbol{z}^{(0)}}$.
3: **for** $l = 1$ **to** $L$ **do**
4:     Let $\Omega^{(l)}$ be a set containing k coordinates of $\boldsymbol{c}^{(l-1)}$ with the largest magnitude. Execute the following updates:

$$y_j^{(l)} = \begin{cases} \frac{z_j^{(l-1)}}{(\boldsymbol{x}^{(l-1)})^T \boldsymbol{z}^{(l-1)}}, & j \in \Omega^{(l)} \\ x_j^{(l-1)}, & j \notin \Omega^{(l)} \end{cases} \tag{2}$$

$$\boldsymbol{z}^{(l)} = \boldsymbol{z}^{(l-1)} + A(\boldsymbol{y}_{\Omega^{(l)}}^{(l)} - \boldsymbol{x}_{\Omega^{(l)}}^{(l-1)}) \tag{3}$$

$$\boldsymbol{z}^{(l)} = \boldsymbol{z}^{(l)}/\|\boldsymbol{y}^{(l)}\|, \quad \boldsymbol{x}^{(l)} = \boldsymbol{y}^{(l)}/\|\boldsymbol{y}^{(l)}\|$$

$$\boldsymbol{c}^{(l)} = \boldsymbol{x}^{(l)} - \frac{\boldsymbol{z}^{(l)}}{(\boldsymbol{x}^{(l-1)})^T \boldsymbol{z}^{(l-1)}}$$

5: **Output:** Approximate dominant eigenvector $\boldsymbol{x}^{(L)}$

---

Algorithm 1 describes our coordinate-wise power method that updates $k$ entries at a time for computing the dominant eigenvector for a symmetric input matrix, while a generalization to asymmetric cases is straightforward. The algorithm starts from an initial vector $\boldsymbol{x}^{(0)}$, and iteratively performs updates $x_i \leftarrow \boldsymbol{a}_i^T \boldsymbol{x}/\boldsymbol{x}^T A \boldsymbol{x}$ with $i$ in a selected set of coordinates $\Omega \subseteq [n]$ defined in step 4, where $\boldsymbol{a}_i$ is the $i$-th row of $A$. The set of indices $\Omega$ is chosen to maximize the difference between the current coordinate value $x_i$ and the next coordinate value $\boldsymbol{a}_i^T \boldsymbol{x}/\boldsymbol{x}^T A \boldsymbol{x}$. $\boldsymbol{z}^{(l)}$ and $\boldsymbol{c}^{(l)}$ are auxiliary vectors. Maintaining $\boldsymbol{z}^{(l)} \equiv A\boldsymbol{x}^{(l)}$ saves much time, while the magnitude of $\boldsymbol{c}$ represents importance of each coordinate and is used to select $\Omega$.

We use the Rayleigh Quotient $\boldsymbol{x}^T A \boldsymbol{x}$ ($\boldsymbol{x}$ is normalized) for scaling, different from $\|A\boldsymbol{x}\|$ in the power method. Our intuition is as follows: on one hand, it is well known that Rayleigh Quotient is the best estimate for eigenvalues. On the other hand, the limit point using $\boldsymbol{x}^T A \boldsymbol{x}$ scaling will satisfy $\bar{\boldsymbol{x}} = A\bar{\boldsymbol{x}}/\bar{\boldsymbol{x}}^T A \bar{\boldsymbol{x}}$, which allows both negative or positive dominant eigenvectors, while the scaling $\|A\boldsymbol{x}\|$ is always positive, so its limit point only lies in the eigenvectors associated with positive eigenvalues, which rules out the possibility of converging to the negative dominant eigenvector.

## 2.1 Coordinate Selection Strategy

An initial understanding for our coordinate selection strategy is that we select coordinates with the largest potential change. With a current iterate $\boldsymbol{x}$ and an arbitrary active set $\Omega$, let $\boldsymbol{y}^\Omega$ be a potential next iterate with only coordinates in $\Omega$ updated, namely,

$$(\boldsymbol{y}^\Omega)_i = \begin{cases} \frac{\boldsymbol{a}_i^T \boldsymbol{x}}{\boldsymbol{x}^T A \boldsymbol{x}}, & i \in \Omega \\ x_i, & i \notin \Omega \end{cases}$$

According to our algorithm, we select active set $\Omega$ to maximize the iterate change. Therefore:

$$\Omega = \underset{I \subset [n], |I|=k}{\arg\max} \left\{ \left\| (\boldsymbol{x} - \frac{A\boldsymbol{x}}{\boldsymbol{x}^T A \boldsymbol{x}})_I \right\|^2 = \|\boldsymbol{y}^I - \boldsymbol{x}\|^2 \right\} = \underset{I \subset [n], |I|=k}{\arg\min} \left\{ \left\| \frac{A\boldsymbol{x}}{\boldsymbol{x}^T A \boldsymbol{x}} - \boldsymbol{y}^I \right\|^2 \overset{\text{def}}{=} \|\boldsymbol{g}\|^2 \right\}$$

This is to say, with our updating rule, our goal of maximizing iteration gap is equivalent to minimizing the difference between the next iterate $\boldsymbol{y}^{(l+1)}$ and $A\boldsymbol{x}^{(l)}/(\boldsymbol{x}^{(l)})^T A \boldsymbol{x}^{(l)}$, where this difference could be interpreted as noise $\boldsymbol{g}^{(l)}$. A good set $\Omega$ ensures a sufficiently small noise $\boldsymbol{g}^{(l)}$, thus achieving a

similar convergence rate in $O(kn)$ time (analyzed later) as the power method does in $O(n^2)$ time. More formal statement for the convergence analysis is given in Section 4.

Another reason for this selection rule is that it incurs little overhead. For each iteration, we maintain a vector $z \equiv Ax$ with $kn$ flops by the updating rule in Eq.(3). And the overhead consists of calculating $c$ and choosing $\Omega$. Both parts cost $O(n)$ operations. Here $\Omega$ is chosen by Hoare's quick selection algorithm [5] to find the $k^{\text{th}}$ largest entry in $|c|$. Thus the overhead is negligible compared with $O(kn)$. Thus CPM spends as much time on each coordinate as PM does on average, while those updated $k$ coordinates are most important. For sparse matrices, the time complexity is $O(n + \frac{k}{n}\text{nnz}(A))$ for each iteration, where $\text{nnz}(A)$ is the number of nonzero elements in matrix $A$.

Although the above analysis gives us a good intuition on how our method works, it doesn't directly show that our coordinate selection strategy has any optimal properties. In next section, we give another interpretation of our coordinate-wise power method and establish its connection with the optimization problem for low-rank approximation.

## 3 Optimization Interpretation

The coordinate descent method [12, 6] was popularized due to its simplicity and good performance. With all but one coordinates fixed, the minimization of the objective function becomes a sequence of subproblems with univariate minimization. When such subproblems are quickly solvable, coordinate descent methods can be efficient. Moreover, in different problem settings, a specific coordinate selecting rule in each iteration makes it possible to further improve the algorithm's efficiency.

The power method reminds us of the rank-one matrix factorization

$$\underset{x \in \mathbb{R}^n, y \in \mathbb{R}^d}{\arg\min} \left\{ f(x, y) = \|A - xy^T\|_F^2 \right\} \tag{4}$$

With alternating minimization, the update for $x$ becomes $x \leftarrow \frac{Ay}{\|y\|^2}$ and vice versa for $y$. Therefore for symmetric matrix, alternating minimization is exactly PM apart from the normalization constant.

Meanwhile, the above similarity between PM and alternating minimization extends to the similarity between CPM and greedy coordinate descent. A more detailed interpretation is in Appendix A.5, where we show the equivalence in the following coordinate selecting rules for Eq.(4): **(a)** largest coordinate value change, denoted as $|\delta x_i|$; **(b)** largest partial gradient (Gauss-Southwell rule), $|\nabla_i f(x)|$; **(c)** largest function value decrease, $|f(x + \delta x_i e_i) - f(x)|$. Therefore, the coordinate selection rule is more formally testified in optimization viewpoint.

### 3.1 Symmetric Greedy Coordinate Descent (SGCD)

We propose an even faster algorithm based on greedy coordinate descent. This method is designed for symmetric matrices and additionally requires to know the sign of the most dominant eigenvalue. We also prove its convergence to the global optimum with a sufficiently close initial point.

A natural alternative objective function specifically for the symmetric case would be

$$\underset{x \in \mathbb{R}^n}{\arg\min} \left\{ f(x) = \|A - xx^T\|_F^2 \right\}. \tag{5}$$

Notice that the stationary points of $f(x)$, which require $\nabla f(x) = 4(\|x\|^2 x - Ax) = 0$, are obtained at eigenvectors: $x_i^* = \sqrt{\lambda_i} v_i$, if the eigenvalue $\lambda_i$ is positive. The global minimum for Eq. (5) is the eigenvector corresponding to the largest positive eigenvalue, not the one with the largest magnitude. For most applications like PageRank we know $\lambda_1$ is positive, but if we want to calculate the negative eigenvalue with the largest magnitude, just optimize on $f = \|A + xx^T\|_F^2$ instead.

Now we introduce Algorithm 2 that optimizes Eq. (5). With coordinate descent, we update the $i$-th coordinate by $x_i^{(l+1)} \leftarrow \arg\min_\alpha f(x^{(l)} + (\alpha - x_i^{(l)})e_i)$, which requires the partial derivative of $f(x)$ in $i$-th coordinate to be zero, i.e.,

$$\nabla_i f(x) = 4(x_i\|x\|_2^2 - a_i^T x) = 0. \tag{6}$$

$$\iff x_i^3 + px_i + q = 0, \text{ where } p = \|x\|^2 - x_i^2 - a_{ii}, \text{ and } q = -a_i^T x + a_{ii}x_i \tag{7}$$

Similar to CPM, the most time consuming part comes from maintaining $z$ ($\equiv Ax$), as the calculation for selecting the criterion $c$ and the coefficient $q$ requires it. Therefore the overall time complexity for one iteration is the same as CPM.

Notice that $c$ from Eq.(6) is the partial gradient of $f$, so we are using the Gauss-Southwell rule to choose the active set. And it is actually the only effective and computationally cheap selection rule among previously analyzed rules **(a), (b)** or **(c)**. For calculating the iterate change $|\delta x_i|$, one needs to obtain roots for $n$ equations. Likewise, the function decrease $|\Delta f_i|$ requires even more work.

Remark: for an unbiased initializer, $\boldsymbol{x}^{(0)}$ should be scaled by a constant $\alpha$ such that

$$\alpha = \arg\min_{a \geq 0} \|A - (a\boldsymbol{x}^{(0)})(a\boldsymbol{x}^{(0)})^T\|_F = \sqrt{\frac{(\boldsymbol{x}^{(0)})^T A \boldsymbol{x}^{(0)}}{\|\boldsymbol{x}^{(0)}\|^4}}$$

---

**Algorithm 2** Symmetric greedy coordinate descent (SGCD)

---

1: **Input:** Symmetric matrix $A \in \mathbb{R}^{n \times n}$, number of selected coordinate, $k$, and number of iterations, $L$.
2: Initialize $\boldsymbol{x}^{(0)} \in \mathbb{R}^n$ and set $\boldsymbol{z}^{(0)} = A\boldsymbol{x}^{(0)}$. Set coordinate selecting criterion $\boldsymbol{c}^{(0)} = \boldsymbol{x}^{(0)} - \frac{\boldsymbol{z}^{(0)}}{\|\boldsymbol{x}^{(0)}\|^2}$.
3: **for** $l = 0$ **to** $L - 1$ **do**
4:    Let $\Omega^{(l)}$ be a set containing $k$ coordinates of $\boldsymbol{c}^{(l)}$ with the largest magnitude. Execute the following updates:

$$
\begin{aligned}
x_j^{(l+1)} &= \begin{cases} \arg\min_\alpha f\left(\boldsymbol{x}^{(l)} + (\alpha - x_j^{(l)})\boldsymbol{e}_j\right), & \text{if } j \in \Omega^{(l)}, \\ x_j^{(l)}, & \text{if } j \notin \Omega^{(l)}. \end{cases} \\
\boldsymbol{z}^{(l+1)} &= \boldsymbol{z}^{(l)} + A(\boldsymbol{x}_{\Omega^{(l)}}^{(l+1)} - \boldsymbol{x}_{\Omega^{(l)}}^{(l)}) \\
\boldsymbol{c}^{(l+1)} &= \boldsymbol{x}^{(l+1)} - \frac{\boldsymbol{z}^{(l+1)}}{\|\boldsymbol{x}^{(l+1)}\|^2}
\end{aligned}
$$

5: **Output:** vector $x^{(L)}$

---

## 4 Convergence Analysis

In the previous section, we propose coordinate-wise power method (CPM) and symmetric greedy coordinate descent (SGCD) on a non-convex function for computing the dominant eigenvector. However, it remains an open problem to prove convergence of coordinate descent methods for general non-convex functions. In this section, we show that both CPM and SGCD converge to the dominant eigenvector under some assumptions.

### 4.1 Convergence of Coordinate-wise Power Method

Consider a positive semidefinite matrix A, and let $\boldsymbol{v}_1$ be its leading eigenvector. For any sequence $(\boldsymbol{x}^{(0)}, \boldsymbol{x}^{(1)}, \cdots)$ generated by Algorithm 1, let $\theta^{(l)}$ to be the angle between vector $\boldsymbol{x}^{(l)}$ and $\boldsymbol{v}_1$, and $\phi^{(l)}(k) \stackrel{\text{def}}{=} \min_{|\Omega|=k} \sqrt{\sum_{i \notin \Omega} (c_i^{(l)})^2}/\|\boldsymbol{c}^{(l)}\|_2 = \|\boldsymbol{g}^{(l)}\|/\|\boldsymbol{c}^{(l)}\|$. The following lemma illustrates convergence of the tangent of $\theta^{(l)}$.

**Lemma 4.1.** *Suppose $k$ is large enough such that*

$$\phi^{(l)}(k) < \frac{\lambda_1 - \lambda_2}{(1 + \tan\theta^{(l)})\lambda_1}. \tag{8}$$

*Then*

$$\tan\theta^{(l+1)} \quad \leq \quad \tan\theta^{(l)}\left(\frac{\lambda_2}{\lambda_1} + \frac{\phi^{(l)}(k))}{\cos\theta^{(l)}}\right) < \tan\theta^{(l)} \tag{9}$$

With the aid of Lemma 4.1, we show the following iteration complexity:

**Theorem 4.2.** *For any sequence $(\boldsymbol{x}^{(0)}, \boldsymbol{x}^{(1)}, \cdots)$ generated by Algorithm 1 with $k$ satisfying $\phi^{(l)}(k) < \frac{\lambda_1 - \lambda_2}{2\lambda_1(1+\tan\theta^{(l)})}$, if $\boldsymbol{x}^{(0)}$ is not orthogonal to $\boldsymbol{v}_1$, then after $T = O(\frac{\lambda_1}{\lambda_1 - \lambda_2}\log(\frac{\tan\theta^{(0)}}{\varepsilon}))$ iterations we have $\tan\theta^{(T)} \leq \varepsilon$.*

The iteration complexity shown is the same as the power method, but since it requires less operations ($O(k\,\text{nnz}(A)/n)$ instead of $O(\text{nnz}(A))$) per iteration, we have

**Corollary 4.2.1.** *If the requirements in Theorem 4.2 apply and additionally $k$ satisfies:*

$$k < n\log((\lambda_1 + \lambda_2)/(2\lambda_1))/\log(\lambda_2/\lambda_1), \tag{10}$$

*CPM has a better convergence rate than PM in terms of the number of equivalent passes over the coordinates.*

The RHS of (10) ranges from $0.06n$ to $0.5n$ when $\frac{\lambda_2}{\lambda_1}$ goes from $10^{-5}$ to $1 - 10^{-5}$. Meanwhile, experiments show that the performance of our algorithms isn't too sensitive to the choice of $k$. Figure 6 in Appendix A.6 illustrates that a sufficiently large range of $k$ guarantees good performances. Thus we use a prescribed $k = \frac{n}{20}$ throughout our experiments in this paper, which saves the burden of tuning parameters and is a theoretically and experimentally favorable choice.

Part of the proof is inspired by the noisy power method [3] in that we consider the unchanged part $g$ as noise. For the sake of a neat proof we require our target matrix to be positive semidefinite, although experimentally a generalization to regular matrices is also valid for our algorithm. Details can be found in Appendix A.1 and A.3.

## 4.2 Local Convergence for Optimization on $\|A - xx^T\|_F^2$

As the objective in Problem (5) is non-convex, it is hard to show global convergence. Clearly, with exact coordinate descent, Algorithm 2 will converge to some stationary point. In the following, we show that Algorithm 2 converges to the global minimum with a starting point sufficiently close to it.

**Theorem 4.3.** *(Local Linear Convergence) For any sequence of iterates $(x^{(0)}, x^{(1)}, \cdots)$ generated by Algorithm 2, assume the starting point $x^{(0)}$ is in a ball centered by $\sqrt{\lambda_1} v_1$ with radius $r = O(\frac{\lambda_1 - \lambda_2}{\sqrt{\lambda_1}})$, or formally, $x^{(0)} \in B_r(\sqrt{\lambda_1} v_1)$, then $(x^0, x^1, \cdots)$ converges to the optima linearly.*

*Specifically, when $k = 1$, then after $T = \frac{14\lambda_1 - 2\lambda_2 + 4 \max_i |a_{ii}|}{\mu} \log \frac{f(x^{(0)}) - f^*}{\varepsilon}$ iterations, we have $f(x^{(T)}) - f^* \leq \varepsilon$, where $f^* = f(\sqrt{\lambda_1} v_1)$ is the global minimum of the objective function $f$, and $\mu = \inf_{x,y \in B_r(\sqrt{\lambda_1} v_1)} \frac{\|\nabla f(x) - \nabla f(y)\|_\infty}{\|x - y\|_1} \in \left[\frac{3(\lambda_1 - \lambda_2)}{n}, 3(\lambda_1 - \lambda_2)\right]$.*

We prove this by showing that the objective (5) is strongly convex and coordinate-wise Lipschitz continuous in a neighborhood of the optimum. The proof is given in Appendix A.4.

Remark: For real-life graphs, the diagonal values $a_{ii} = 0$, and the coefficient in the iteration complexity could be simplified as $\frac{14\lambda_1 - 2\lambda_2}{\mu}$ when $k = 1$.

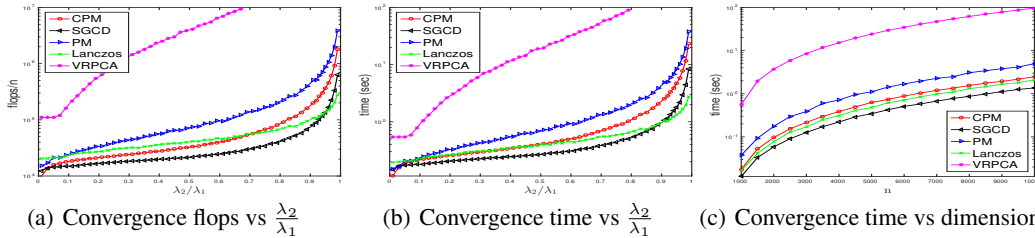

(a) Convergence flops vs $\frac{\lambda_2}{\lambda_1}$     (b) Convergence time vs $\frac{\lambda_2}{\lambda_1}$     (c) Convergence time vs dimension

Figure 2: **Matrix properties affecting performance.** Figure 2(a), 2(b) show the performance of five methods with $\frac{\lambda_2}{\lambda_1}$ ranging from 0.01 to 0.99 and fixed matrix size $n = 5000$. In Figure 2(a) the measurement is FLOPs while in Figure 2(b) Y-axis is CPU time. Figure 2(c) shows how the convergence time varies with the dimension when fixing $\frac{\lambda_2}{\lambda_1} = 2/3$. In all figures Y-axis is in log scale for better observation. Results are averaged over from 20 runs.

## 5 Experiments

In this section, we compared our algorithms with PM, Lanczos method [14], and VRPCA [16] on dense as well as sparse dataset. All the experiments were executed on Intel(R) Xeon(R) E5430 machine with 16G RAM and Linux OS. We implement all the five algorithms in C++ with Eigen library.

### 5.1 Comparison on Dense and Simulated Dataset

We compare PM with our CPM and SGCD methods to show how coordinate-wise mechanism improves the original method. Further, we compared with a state-of-the-art algorithm *Lanczos* method. Besides, we also include a recent proposed stochastic SVD algorithm, *VRPCA*, that enjoys exponential convergence rate and shows similar insight in viewing the data in a separable way.

With dense and synthetic matrices, we are able to test the condition that our methods are preferable, and how the properties of the matrix, like $\lambda_2/\lambda_1$ or the dimension, affect the performance. For each algorithm, we start from the same random vector, and set stopping condition to be $\cos \theta \geq 1 - \epsilon, \epsilon = 10^{-6}$, where $\theta$ is the angle between the current iterate and the dominant eigenvector.

First we compare the performances with number of FLOPs (Floating Point Operations), which could better illustrate how greediness affects the algorithm's efficiency. From Figure 2(a) we can see our method shows much better performance than PM, especially when $\lambda_2/\lambda_1 \to 1$, where CPM and SGCD respectively achieve more than 2 and 3 times faster than PM. Figure 2(b) shows running time using five methods under different eigenvalue ratios $\lambda_2/\lambda_1$. We can see that only in some extreme cases when PM converges in less than 0.1 second, PM is comparable to our methods. In Figure 2(c) the testing factor is the dimension, which shows the performance is independent of the size of $n$. Meanwhile, in most cases, SGCD is better than Lanczos method. And although VRPCA has better convergence rate, it requires at least $10n^2$ operations for one data pass. Therefore in real applications, it is not even comparable to PM.

## 5.2 Comparison on Sparse and Real Dataset

Table 1: Six datasets and the performance of three methods on them.

| Dataset | n | nnz($A$) | nnz/n | $\frac{\lambda_2}{\lambda_1}$ | Time (sec) | | | | |
|---------|---|----------|-------|-------------------------------|------|------|------|---------|-------|
| | | | | | PM | CPM | SGCD | Lanczos | VRPCA |
| com-Orkut | 3.07M | 234M | 76.3 | 0.71 | 109.6 | 31.5 | **19.3** | 63.6 | 189.7 |
| soc-LiveJournal | 4.85M | 86M | 17.8 | 0.78 | 58.5 | 17.9 | **13.7** | 25.8 | 88.1 |
| soc-Pokec | 1.63M | 44M | 27.3 | 0.95 | 118 | 26.5 | **5.2** | 14.2 | 596.2 |
| web-Stanford | 282K | 3.99M | 14.1 | 0.95 | 8.15 | 1.05 | **0.54** | 0.69 | 7.55 |
| ego-Gplus | 108K | 30.5M | 283 | 0.51 | 0.99 | **0.57** | 0.61 | 1.01 | 5.06 |
| ego-Twitter | 81.3K | 2.68M | 33 | 0.65 | 0.31 | 0.15 | **0.11** | 0.19 | 0.98 |

To test the scalability of our methods, we further test and compare our methods on large and sparse datasets. We use the following real datasets:

*1)* com-Orkut: Orkut online social network
*2)* soc-LiveJournal: On-line community for maintaining journals, individual and group blogs
*3)* soc-Pokec: Pokec, most popular on-line social network in Slovakia
*4)* web-Stanford: Pages from Stanford University (stanford.edu) and hyperlinks between them
*5)* ego-Gplus (Google+): Social circles from Google+
*6)* ego-Twitter: Social circles from Twitter

The statistics of the datasets are summarized in Table 1, which includes the essential properties of the datasets that affect the performances and the average CPU time for reaching $\cos\theta_{\boldsymbol{x},\boldsymbol{v}_1} \geq 1 - 10^{-6}$. Figure 3 shows $\tan\theta_{\boldsymbol{x},\boldsymbol{v}_1}$ against the CPU time for the four methods with multiple datasets.

From the statistics in Table 1 we can see that in all the cases, either CPM or SGCD performs the best. CPM is roughly 2-8 times faster than PM, while SGCD reaches up to 23 times and 3 times faster than PM and Lanczos method respectively. Our methods show their privilege in the soc-Pokec(3(c)) and web-Stanford(3(d)), the most ill-conditioned cases ($\lambda_2/\lambda_1 \approx 0.95$), achieving 15 or 23 times of speedup on PM with SGCD. Meanwhile, when the condition number of the datasets is not too small (see 3(a),3(b),3(e),3(f)), both CPM and SGCD outperform PM as well as Lanczos method. And

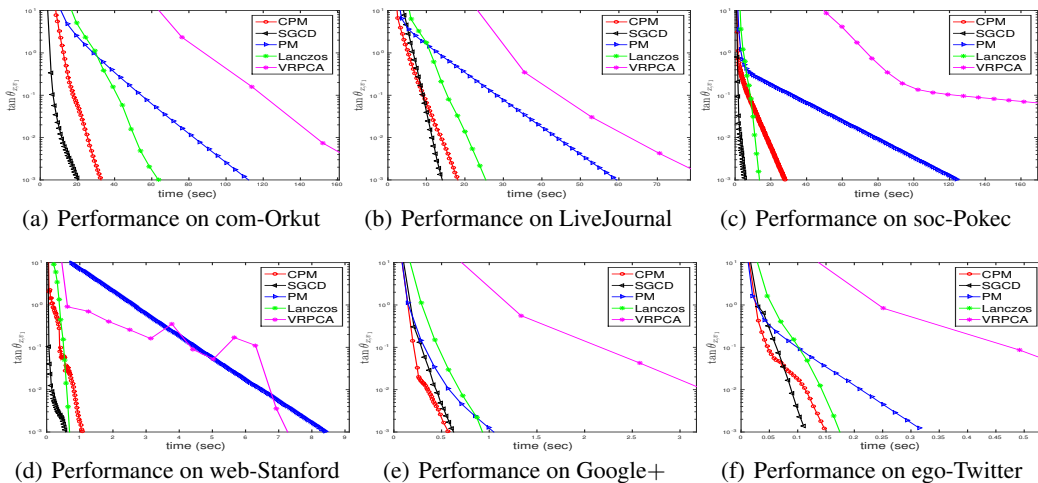

(a) Performance on com-Orkut  (b) Performance on LiveJournal  (c) Performance on soc-Pokec

(d) Performance on web-Stanford  (e) Performance on Google+  (f) Performance on ego-Twitter

Figure 3: **Time comparison for sparse dataset.** X-axis shows the CPU time while Y-axis is log scaled $\tan\theta$ between $\boldsymbol{x}$ and $\boldsymbol{v}_1$. The empirical performance shows all three methods have linear convergence.

similar to the reasoning in the dense case, although VRPCA requires less iterations for convergence, the overall CPU time is much longer than others in practice.

In summary of performances on both dense and sparse datasets, SGCD is the fastest among others.

# 6 Other Application and Extensions

## 6.1 Comparison on Out-of-core Real Dataset

An important application for coordinate-wise power method is the case when data can not fit into memory. Existing methods can't be easily applied to out-of-core dataset. Most existing methods don't indicate how we can update part of the coordinates multiple times and fully reuse part of the matrix corresponding to those active coordinates. Therefore the data loading and data processing time are highly unbalanced. A naive way of using PM would be repetitively loading part of the matrix from the disk and calculating that part of matrix-vector multiplication. But from Figure 4 we can see reading from the disk costs much more time than the process of computation, therefore we will waste a lot of time if we cannot fully use the data before dumping it. For CPM, as we showed in Theorem 4.1 that updating only $k$ coordinates of iterate $x$ may still enhance the target direction, we could do

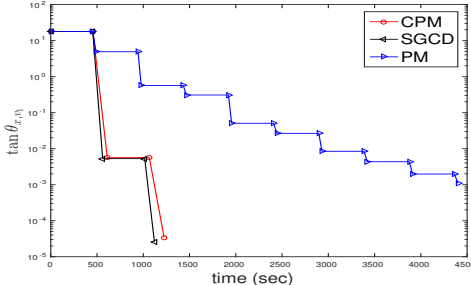

Figure 4: **A pseudograph for time comparison of out-of-core dataset from Twitter.** Each "staircase" illustrates the performance of one data pass. The flat part indicates the stage of loading data, while the downward part shows the phase of processing data. As we only updated auxiliary vectors instead of the iterate every time we load part of the matrix, we could not test performances until a whole data pass. Therefore for the sake of clear observation, we group together the loading phase and the processing phase in each data pass.

matrix vector multiplication multiple times after one single loading. As with SGCD, optimization on part of $x$ for several times will also decrease the function value.

We did experiments on the dataset from Twitter [10] using out-of-core version of the three algorithms shown in Algorithm 3 in Appendix A.7. The data, which contains 41.7 million user profiles and 1.47 billion social relations, is originally 25.6 GB and then separated into 5 files. In Figure 4, we can see that after data pass, our methods can already reach rather high precision, which compresses hours of processing time to 8 minutes.

## 6.2 Extension to other linear algebraic methods

With the interpretation in optimization, we could apply a coordinate-wise mechanism to PM and get good performance. Meanwhile, for some other iterative methods in linear algebra, if the connection to optimization is valid, or if the update is separable for each coordinate, the coordinate-wise mechanism may also be applicable, like Jacobi method.

For diagonal dominant matrices, Jacobi iteration [15] is a classical method for solving linear system $Ax = b$ with linear convergence rate. The iteration procedure is:

**Initialize:** $A \to D + R$, where $D =$Diag$(A)$, and $R = A - D$.
**Iterations:** $x^+ \leftarrow D^{-1}(b - Rx)$.

This method is similar to the vanilla power method, which includes a matrix vector multiplication $-Rx$ with an extra translation $b$ and a normalization step $D^{-1}$. Therefore, a potential similar realization of greedy coordinate-wise mechanism is also applicable here. See Appendix A.8 for more experiments and analyses, where we also specify its relation to Gauss-Seidel iteration [15].

# 7 Conclusion

In summary, we propose a new coordinate-wise power method and greedy coordinate descent method for computing the most dominant eigenvector of a matrix. This problem is critical to many applications in machine learning. Our methods have convergence guarantees and achieve up to 23 times of speedup on both real and synthetic data, as compared to the vanilla power method.

## Acknowledgements

This research was supported by NSF grants CCF-1320746, IIS-1546452 and CCF-1564000.

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
