[Supplementary Material]

# A Appendix

## A.1 Proof of Lemma 4.1

We consider the difference between $y^{(l+1)}$ and $\frac{Ax^{(l)}}{(x^{(l)})^T Ax^{(l)}}$ as noise, denoted by $g^{(l)}$. To prove the results, we need to use Lemma A.1:

**Lemma A.1.** *For any unit norm $x \in \mathbb{R}^n$, $c \overset{def}{=} x - \frac{Ax}{x^T Ax}$ satisfies $x^T Ax \|c\| \le \sin\theta\lambda_1$, where $\theta$ is the angle between $v_1$ and $x$.*

*Proof.* Write $x = \cos\theta v_1 + \sin\theta u$, where $u \perp v_1$. Then

$$
\begin{aligned}
&\|Ax - (x^T Ax)x\|^2 \\
=\ & \|\cos\theta\lambda_1 v_1 + \sin\theta Au - (\cos^2\theta\lambda_1 + \sin^2\theta u^T Au)(\cos\theta v_1 + \sin\theta u)\|^2 \\
=\ & \|\cos\theta\sin^2\theta(\lambda_1 - u^T Au)v_1 + \sin\theta(\cos^2\theta((u^T Au)u - \lambda_1 u) + (Au - (u^T Au)u))\|^2
\end{aligned}
$$

Notice $u$, $Au - (u^T Au)u$ and $v_1$ are orthogonal to each other. Therefore,

$$
\begin{aligned}
&\|Ax - (x^T Ax)x\|^2 \\
=\ & \cos^2\theta\sin^2\theta(\lambda_1 - u^T Au)^2 + \sin^2\theta\|Au - (u^T Au)u\|^2 \\
\le\ & \sin^2\theta(\lambda_1 - u^T Au)^2 + \sin^2\theta\|Au\|^2 \\
\le\ & (\lambda_1\sin\theta)^2
\end{aligned}
$$

The last step makes use of the fact that $\lambda_1 A - A^T A$ is positive semidefinite, so that $\lambda_1 u^T Au \ge u^T A^T Au = \|Au\|^2$ for any $u$. $\qquad\square$

Now we have the following corollary.

**Corollary A.1.1.** *For $g^{(l)}$, $x^{(l)}$, $\phi^{(l)}(k)$ defined for Algorithm 1, $(x^{(l)})^T Ax^{(l)}\|g^{(l)}\| \le \sin\theta^{(l)}\lambda_1\phi^{(l)}(k)$.*

This result is crucial to the following proof of Lemma 4.1.

Figure 5: The central right triangle has a base-side of length $\lambda_1\cos\theta^{(l)}$ and height of at most $\lambda_2\sin\theta^{(l)}$. The dashed line that ends in the center of the circle is $Ax$ and the straight lines with an arrow are possible $g$'s. Then Eq (11) can be represented by the tangent of the angle between the base-side and the dotted lines that ends on the circle of radius $x^T Ax\|g\|$. Therefore $\tan\theta^{(l+1)} \le \frac{\lambda_2\sin\theta^l + x^T Ax\|g\|/\cos\theta^{(l+1)}}{\lambda_1\cos\theta^{(l)}}$.

Let $U \in \mathbb{R}^{n \times (n-1)} = [\boldsymbol{v}_2 | \boldsymbol{v}_3 | \cdots | \boldsymbol{v}_n]$ denote the orthonormal space of $\boldsymbol{v}_1$. The next iterate satisfies:

$$
\begin{aligned}
& \tan \theta^{(l+1)} \\
= & \frac{\|U^T \boldsymbol{y}^{(l+1)}\|}{\boldsymbol{v}_1^T \cdot \boldsymbol{y}^{(l+1)}} \\
= & \frac{\|U^T \frac{A\boldsymbol{x}^{(l)}}{(\boldsymbol{x}^{(l)})^T A \boldsymbol{x}^{(l)}} + U^T \boldsymbol{g}^{(l)}\|}{\boldsymbol{v}_1^T \frac{A\boldsymbol{x}^{(l)}}{(\boldsymbol{x}^{(l)})^T A \boldsymbol{x}^{(l)}} + \boldsymbol{v}_1^T \boldsymbol{g}^{(l)}} \\
\leq & \frac{\sin \theta^{(l)} \lambda_2 + (\boldsymbol{x}^{(l)})^T A \boldsymbol{x}^{(l)} \| U^T \boldsymbol{g}^{(l)}\|}{\cos \theta^{(l)} \lambda_1 + (\boldsymbol{x}^{(l)})^T A \boldsymbol{x}^{(l)} \boldsymbol{v}_1^T \boldsymbol{g}^{(l)}} \qquad (11) \\
\leq & \frac{\sin \theta^{(l)} \lambda_2 + (\boldsymbol{x}^{(l)})^T A \boldsymbol{x}^{(l)} \| \boldsymbol{g}^{(l)}\| / \cos \theta^{(l+1)}}{\cos \theta^{(l)} \lambda_1} \qquad (12)
\end{aligned}
$$

The logic from Eq (11) to Eq (12) is interpreted in Figure A.1.

Applying Lemma A.1 on Inequality (11), one gets

$$
\tan \theta^{(l+1)} \quad \leq \quad \frac{\sin \theta^{(l)} \lambda_2 + \phi(k) \sin \theta^{(l)} \lambda_1}{\cos \theta^{(l)} \lambda_1 - \phi(k) \sin \theta^{(l)} \lambda_1} = \tan \theta^{(l)} \frac{\lambda_2 + \lambda_1 \phi(k)}{\lambda_1 (1 - \tan \theta^{(l)} \phi(k))}
$$

Therefore with large enough $k$ such that $\phi(k) \leq \frac{\lambda_1 - \lambda_2}{\lambda_1 (1 + \tan \theta^{(l)})}$, we could guarentee that $\theta^{(l+1)} < \theta^{(l)}$, $\frac{1}{\cos \theta^{(l+1)}} < \frac{1}{\cos \theta^{(l)}}$. So continuing Eq. (12), we have

$$
\begin{aligned}
\tan \theta^{(l+1)} \quad \leq & \quad \frac{\sin \theta^{(l)} \lambda_2 + (\boldsymbol{x}^{(l)})^T A \boldsymbol{x}^{(l)} \| \boldsymbol{g}^{(l)}\| / \cos \theta^{(l)}}{\cos \theta^{(l)} \lambda_1} \\
\leq & \quad \frac{\sin \theta^{(l)} \lambda_2 + \phi(k) \sin \theta^{(l)} \lambda_1 / \cos \theta^{(l)}}{\cos \theta^{(l)} \lambda_1} \\
= & \quad \tan \theta^{(l)} \left( \frac{\lambda_2}{\lambda_1} + \frac{\phi(k)}{\cos \theta^{(l)}} \right)
\end{aligned}
$$

## A.2 Proof of Theorem 4.2

When $\phi^{(l)}(k) \leq (\lambda_1 - \lambda_2)/(2\lambda_1(1 + \tan \theta^{(l)}))$, we obtain that,

$$
\frac{\phi^{(l)}(k)}{\cos \theta^{(l)}} \quad \leq \quad \frac{\lambda_1 - \lambda_2}{2\lambda_1 (\cos \theta^{(l)} + \sin \theta^{(l)})} \leq (\lambda_1 - \lambda_2)/(2\lambda_1)
$$

$$
\begin{aligned}
\tan \theta^{(l)} \quad \leq & \quad \tan \theta^{(l-1)} (\lambda_2/\lambda_1 + (\lambda_1 - \lambda_2)/(2\lambda_1)) \qquad (13) \\
\leq & \quad \tan \theta^{(0)} \left( \frac{\lambda_1 + \lambda_2}{2\lambda_1} \right)^l \qquad (14) \\
\leq & \quad \tan \theta^{(0)} e^{-l(\lambda_1 - \lambda_2)/(2\lambda_1)} \qquad (15)
\end{aligned}
$$

Therefore when $l \geq 2\frac{\lambda_1}{\lambda_1 - \lambda_2} \log \frac{\tan \theta^{(0)}}{\varepsilon}$, $\tan \theta^{(l)} \leq \varepsilon$.

## A.3 Proof of Corollary

To compare convergence rate between CPM and PM in comparable operations, one should notice one iteration of CPM costs around $\frac{k}{n}$ percentage of operations as PM does. Therefore we should compare our convergence rate $\frac{\lambda_1 + \lambda_2}{2\lambda_1}$ with $\left( \frac{\lambda_2}{\lambda_1} \right)^{\frac{k}{n}}$. Therefore when

$$
k < \frac{\log \left( \frac{\lambda_1 + \lambda_2}{2\lambda_1} \right)}{\log \frac{\lambda_2}{\lambda_1}} n,
$$

our convergence rate is better than power method in terms of equivalent passes over data.

## A.4 Proof of Theorem 4.3

**Lemma A.2.** *In objective (5) $f(\boldsymbol{x}) = \|A - \boldsymbol{x}\boldsymbol{x}^T\|_F^2$, it can be shown that within area $x \in B_r(\sqrt{\lambda_1}\boldsymbol{v}_1) = \{y | \|y - \sqrt{\lambda_1}\boldsymbol{v}_1\| \le r\}, r = O(\sqrt{\lambda_1} - \frac{\lambda_2}{\sqrt{\lambda_1}}), f(\boldsymbol{x})$ is strongly convex.*

**Proof of A.2.** Notice for the objective function $f$, $\nabla f(\boldsymbol{x}) = -4(A\boldsymbol{x} - \|\boldsymbol{x}\|^2 \boldsymbol{x})$, Hessian matrix $H(\boldsymbol{x}) = -4(A - \|\boldsymbol{x}\|^2 I - 2\boldsymbol{x}\boldsymbol{x}^T)$, and its stationary points are $x_i = \sqrt{\lambda_i}\boldsymbol{v}_i$. Denote the eigenvalues $\lambda_1 > \lambda_2 \ge \cdots \lambda_r \ge 0 > \cdots \lambda_n$, and with the assumption that the dominant eigenvalue is positive, we have $\lambda_1 > |\lambda_n|$.

At point $\sqrt{\lambda_1}\boldsymbol{v}_1$, the Hessian matrix of $f$ is positive definite:

$$
\begin{aligned}
H(\sqrt{\lambda_1}\boldsymbol{v}_1) &= -4(A - \lambda_1 I - 2\lambda_1 \boldsymbol{v}_1 \boldsymbol{v}_1^T) \\
&= 4\lambda_1 \boldsymbol{v}_1 \boldsymbol{v}_1^T + 4\lambda_1 I - 4\sum_{i=2}^{n} \lambda_i \boldsymbol{v}_i \boldsymbol{v}_i^T
\end{aligned}
$$

Therefore, $H$ has the same eigenvectors as A: $\boldsymbol{v}_1, \boldsymbol{v}_2 \cdots \boldsymbol{v}_n$, with respect to eigenvalues $8\lambda_1, 4(\lambda_1 - \lambda_2), 4(\lambda_1 - \lambda_3), \cdots, 4(\lambda_1 - \lambda_n)$, which indicates that $H$ is positive definite with its smallest eigenvector $4(\lambda_1 - \lambda_2) > 0$.

Now to show $f$ is strongly convex within the neighborhood $B_r(\sqrt{\lambda_1}\boldsymbol{v}_1)$, we denote $\boldsymbol{x} = \sqrt{\lambda_1}\boldsymbol{v}_1 + \boldsymbol{h}, \|\boldsymbol{h}\| \le r$, and introduce

$$
G(\boldsymbol{h}, \boldsymbol{g}) \overset{\text{def}}{=\!=} \frac{\boldsymbol{g}^T H(\sqrt{\lambda_1}\boldsymbol{v}_1 + \boldsymbol{h})\boldsymbol{g}}{\boldsymbol{g}^T \boldsymbol{g}}
$$

which could represent the range of eigenvalues to $H(\sqrt{\lambda_1}\boldsymbol{v}_1 + \boldsymbol{h})$. Notice

$$
\begin{aligned}
\nabla_{\boldsymbol{h}} G(\boldsymbol{h}, \boldsymbol{g}) &= 8\sqrt{\lambda_1}\boldsymbol{v}_1^T + 8\boldsymbol{h} + 16(\sqrt{\lambda_1}\boldsymbol{v}_1^T \frac{\boldsymbol{g}}{\|\boldsymbol{g}\|} + \boldsymbol{h}^T \frac{\boldsymbol{g}}{\|\boldsymbol{g}\|})\frac{\boldsymbol{g}}{\|\boldsymbol{g}\|} \\
, \text{and } \|\nabla_{\boldsymbol{h}} G(\boldsymbol{h}, \boldsymbol{g})\| &\le 8\sqrt{\lambda_1} + 8\|\boldsymbol{h}\| + 16(\sqrt{\lambda_1} + \|\boldsymbol{h}\|) \\
&= 24(\sqrt{\lambda_1} + \|\boldsymbol{h}\|) \\
&\le 24(\sqrt{\lambda_1} + r), \forall \boldsymbol{h} \in B_r(0)
\end{aligned}
$$

By mean-value theorem,

$$
\begin{aligned}
|G(\boldsymbol{h}, \boldsymbol{g}) - G(0, \boldsymbol{g})| &\le (\sup_{\boldsymbol{h} \in B_r(0)} \|\nabla_{\boldsymbol{h}} G(\boldsymbol{h}, \boldsymbol{g})\|)\|\boldsymbol{h}\| \\
&\le 24(\sqrt{\lambda_1} + r)r, \forall \boldsymbol{h} \in B_r(0), \forall \boldsymbol{g} \in \mathbb{R}^n
\end{aligned}
$$

With some proper relaxation, when $r = \frac{1}{30}\frac{\lambda_1 - \lambda_2}{\sqrt{\lambda_1}}$, we have $|G(\boldsymbol{h}, \boldsymbol{g}) - G(0, \boldsymbol{g})| \le \lambda_1 - \lambda_2$.

Recall $G(0, \boldsymbol{g}) = \frac{\boldsymbol{g}^T H(\sqrt{\lambda_1}\boldsymbol{v}_1)\boldsymbol{g}}{\|\boldsymbol{g}\|^2} \ge 4(\lambda_1 - \lambda_2), \forall \boldsymbol{g} \in \mathbb{R}^n$.

$$
\begin{aligned}
G(\boldsymbol{h}, \boldsymbol{g}) &\ge G(0, \boldsymbol{g}) - |G(\boldsymbol{h}, \boldsymbol{g}) - G(0, \boldsymbol{g})| \\
&\ge 3(\lambda_1 - \lambda_2), \forall \boldsymbol{g} \in \mathbb{R}^n, \|\boldsymbol{h}\| < r, \\
i.e. \\
H(\sqrt{\lambda_1}\boldsymbol{v}_1 + \boldsymbol{h}) &\succeq 3(\lambda_1 - \lambda_2), \forall \boldsymbol{h}, \|\boldsymbol{h}\| \le \frac{1}{30}\frac{\lambda_1 - \lambda_2}{\sqrt{\lambda_1}}
\end{aligned}
$$

Therefore the cost function is $3(\lambda_1 - \lambda_2)$-strongly convex within the area $\boldsymbol{x} \in B_r(\sqrt{\lambda_1}\boldsymbol{v}_1)$.  □

**Lemma A.3.** *In area $B_r(\sqrt{\lambda_1}\boldsymbol{v}_1)$, where $r = \frac{\lambda_1 - \lambda_2}{30\sqrt{\lambda_1}}$, $\nabla_i f$ satisfies coordinate-wise Lipschitz continuous with parameter $L \le 14\lambda_1 - 2\lambda_2 + 4\max_i |a_{ii}|$.*

**Proof of Lemma A.3:** Our goal is to find $L$ that satisfies $|\nabla_i f(\boldsymbol{x} + \alpha \boldsymbol{e}_i) - \nabla_i f(\boldsymbol{x})| \le L|\alpha|$, $\forall \boldsymbol{x}, \alpha$ s.t. $\boldsymbol{x}, \boldsymbol{x} + \alpha \boldsymbol{e}_i \in B_r(\sqrt{\lambda_1}\boldsymbol{v}_1)$.

Notice that $r = \frac{\lambda_1 - \lambda_2}{30\sqrt{\lambda_1}}$, and $\|\boldsymbol{x}\| \le \sqrt{\lambda_1} + r$, $|\alpha| \le 2r$.
Now

$$
\begin{aligned}
& |\nabla_i f(\boldsymbol{x} + \alpha \boldsymbol{e}_i) - \nabla_i f(\boldsymbol{x})| \\
=\ & 4|\|\boldsymbol{x} + \alpha \boldsymbol{e}_i\|^2 (x_i + \alpha) - a_{ii}\alpha - \|\boldsymbol{x}\|^2 x_i| \\
\le\ & 4|\|\boldsymbol{x} + \alpha \boldsymbol{e}_i\|^2 \alpha + \alpha^2 x_i + 2\alpha x_i^2| + 4|a_{ii}\alpha| \\
\le\ & 4|\alpha|((\sqrt{\lambda_1} + r)^2 + 2r(\sqrt{\lambda_1} + r) + 2(\sqrt{\lambda_1} + r)^2) + 4|a_{ii}\alpha| \\
=\ & 4|\alpha|(3\lambda_1 + 10\sqrt{\lambda_1}r + 5r^2) + 4|a_{ii}\alpha| \\
\le\ & [12\lambda_1 + 2(\lambda_1 - \lambda_2) + 4|a_{ii}|]|\alpha|
\end{aligned}
$$

$\square$

Remark: $L = 14\lambda_1 - 2\lambda_2 + 4\max_i |a_{ii}|$, for real application like social network, $a_{ii} = 0$ and $L = 14\lambda_1 - 2\lambda_2$.

With the Lipschitz continuous and strongly convex properties, we show convergence by quoting the result of [13]:

**Lemma A.4.** *When $f$ is strongly convex as $\nabla^2 f \succeq \mu I$, and $\nabla f$ satisfies coordinate-wise $L$-Lipschitz continuous, meaning*

$$
|\nabla_i f(\boldsymbol{x} + \alpha \boldsymbol{e}_i) - \nabla_i f(\boldsymbol{x})| \le L|\alpha|,
$$

*$\forall i = 1, 2, \cdots n, \forall \boldsymbol{x} \in$ convex set $\mathbb{S}$, and $\forall \alpha$ such that $\boldsymbol{x} + \alpha \in \mathbb{S}$. Then with Gauss-Southwell rule the optimization on $f$ satisfies linear convergence:*

$$
f(\boldsymbol{x}^{(l+1)}) - f(\boldsymbol{x}^*) \le (1 - \frac{\mu_1}{L})[f(\boldsymbol{x}^{(l)}) - f(\boldsymbol{x}^*)]. \tag{16}
$$

*Here $\mu_1 = \inf_{\boldsymbol{x}, \boldsymbol{y} \in \mathbb{S}} \frac{\|\nabla f(\boldsymbol{x}) - \nabla f(\boldsymbol{y})\|_\infty}{\|\boldsymbol{x} - \boldsymbol{y}\|_1} \in [\frac{\mu}{n}, \mu]$*

Therefore, the convergence rate for updating one coordinate at a time with Gauss-Southwell rule becomes $(1 - \frac{\mu}{L})^n$, $\mu = \inf_{\boldsymbol{x}, \boldsymbol{y}} \frac{\|\nabla f(\boldsymbol{x}) - \nabla f(\boldsymbol{y})\|_\infty}{\|\boldsymbol{x} - \boldsymbol{y}\|_1} \in [\frac{3(\lambda_1 - \lambda_2)}{n}, 3(\lambda_1 - \lambda_2)]$, $L = 14\lambda_1 - 2\lambda_2 + 4\max_i |a_{ii}|$.

## A.5 Greedy Coordinate Descent and Coordinate Selection Rules

For an arbitrary matrix $A \in \mathbb{R}^{n \times d}$, we can formulate rank-1 matrix approximation:

$$
\underset{\boldsymbol{x} \in \mathbb{R}^n, \boldsymbol{y} \in \mathbb{R}^d}{\arg\min}\ f(\boldsymbol{x}, \boldsymbol{y}) = \|A - \boldsymbol{x}\boldsymbol{y}^T\|_F^2 \tag{17}
$$

Notice that $\nabla_{\boldsymbol{x}} f(\boldsymbol{x}, \boldsymbol{y}) = 2(\|\boldsymbol{y}\|^2 \boldsymbol{x} - A\boldsymbol{y})$. When fixing $\boldsymbol{y}$, we obtain the optimal solution of $\boldsymbol{x}$ to be $\boldsymbol{x} = \frac{A\boldsymbol{y}}{\|\boldsymbol{y}\|^2}$ and vice versa, $\boldsymbol{y} = \frac{A^T \boldsymbol{x}}{\|\boldsymbol{x}\|^2}$. And for symmetric matrices, this alternating minimization algorithm is exactly power method apart from the normalization constant.

Recall our coordinate-wise power method. At each iteration we only update the coordinates with the largest changes. Nevertheless here we can formally interpret this rule as the well-studied Gauss-Southwell rule [12], where the coordinates that maximize the gradient norm is selected. As $\nabla_{x_i} f(\boldsymbol{x}, \boldsymbol{y}) = 2(\|\boldsymbol{y}\|^2 x_i - \boldsymbol{a}_i^T \boldsymbol{y}) = 2\|\boldsymbol{y}\|^2 (x_i - \frac{\boldsymbol{a}_i^T \boldsymbol{y}}{\|\boldsymbol{y}\|^2})$, Gauss-Southwell gives the same choice of coordinates as our coordinate-wise power method.

Meanwhile, specifically for quadratic objectives, Gauss-Southwell rule actually select the coordinates based on the decrease in the objective function, leading to optimal updates, i.e.,

$$
\Delta f_i \ :=\ f(\boldsymbol{x}', \boldsymbol{y}) - f(\boldsymbol{x}, \boldsymbol{y}) = -\|\boldsymbol{y}\|_2^2 (x_i - \frac{\boldsymbol{a}_i^T \boldsymbol{y}}{\|\boldsymbol{y}\|_2^2})^2 = -\frac{(\nabla_{x_i} f)^2}{4\|\boldsymbol{y}\|^2}
$$

where $\boldsymbol{x}' = \boldsymbol{x} + (x_i' - x_i)\boldsymbol{e}_i$, and $x_i' = \frac{\boldsymbol{a}_i^T \boldsymbol{y}}{\|\boldsymbol{y}\|^2}$ is the updated coordinate.

Here we summarize the three coordinate selection rules: **(a)** largest coordinate value change, $|x_i' - x_i|$, where $x_i'$ is the next iterate; **(b)** largest partial gradient (Gauss-Southwell), $|\nabla_i f(\boldsymbol{x})|$; **(c)** largest

function value decrease, $|f(\boldsymbol{x}') - f(\boldsymbol{x})|$, where $\boldsymbol{x}'_i = \boldsymbol{x} + (x'_i - x_i)\boldsymbol{e}_i$. With the good property of quadratic function Eq. (4), for each alternating minimization step, the three selection rules are equivalent. Therefore now with the aid of the objective function, our coordinate selection strategy in CPM, similar as in **(a)**, is now consistent with the rule **(c)** with its nature in choosing the most "important" coordinates.

Given the optimization interpretation, the extension of CPM to computing the top-$r$ eigenvectors of a symmetric matrix is straightforward. For the objective function $f(X, Y) = \|A - XY^T\|_F^2$, where $X, Y \in \mathbb{R}^{n \times r}$, the partial gradient of $f(X, Y)$ with respect to matrices $X, Y$ becomes $2(XY^TY - AY)$ and $2(YX^TX - AX)$. By evaluating the norm in each rows of the gradient, we could select and update row by row for $X$ and $Y$ by $\boldsymbol{a}_i^T Y(Y^TY)^{-1}$ and $\boldsymbol{a}_i^T X(X^TX)^{-1}$. Although the algorithm is well-defined and can speedup power method for computing top-$r$ eigenvectors, power method (a.k.a. subspace iteration) is typically not used for computing the dominant $r$(especially for large $r$) eigenvectors [16]. Therefore we don't expand the discussion of this direction here.

### A.6 Choice of $k$

The choice of $k$ could be viewed as choosing the block size for greedy block coordinate descent, which is usually tuned in practice or determined by objective's separable property.

However, it would be better if $k$ could be prescribed and only depend on $n$, as we don't know other properties like $\frac{\lambda_2}{\lambda_1}$ beforehand. In Corollary 4.2.1 it shows the upper bound of $k$ ranges from $6\%n$ to $50\%n$ when $\frac{\lambda_2}{\lambda_1}$ ranges from $10^{-5}$ to $1 - 10^{-5}$. Meanwhile, experiments also show that the performance of our algorithms isn't too sensitive to the choice of $k$. See Figure 6 a large range of $k$ guarantees good performances. Thus we chose $k = \frac{n}{20}$ throught out experiments in this paper, which is a theoretically and experimentally favorable choice.

Figure 6: Convergence time with different $k$ for different $\lambda_2/\lambda_1$

### A.7 Out-of-core Algorithm

Here we formally present the algorithm for the out-of-core case.

---
**Algorithm 3** PM,CPM,SGCD for out-of-core matrix $A$

---
1: **Initialization:** Separate and save matrix $A \in \mathbb{R}^{n \times n}$ into $m$ files, each containing $n/m$ rows of $A$ and being able to fit into memory. Initialize random unit vector $x^{(0)}$.
2: **for** $l = 1$ **to** $L$ **do**
3:     **for** $i = 1$ **to** m **do**
4:         Set $\Omega = (\frac{(i-1)n}{m} + 1) : \frac{in}{m}$.
5:         For PM, calculate $A_{\Omega,:}\boldsymbol{x}^{(l-1)}$
6:         For CPM, do Step 4 in Algorithm 1 for $t$ times.
7:         For SGCD, do Step 4 in Algorithm 2 for $t$ times.
8:     Update $\boldsymbol{x}^{(l)}$.
9: **Output:** Approximate dominant eigenvector $\boldsymbol{x}^{(L)}$

---

### A.8 Extension of Coordinate-wise Mechanism on the Jacobi method

For coordinate-wise Jacobi method for solving $A\boldsymbol{x} = \boldsymbol{b}$, the algorithm is included here:

And for each iteration, it takes $O(nnz(R) + n)$ operations for naive Jacobi, and $O(\frac{k}{n}nnz(R) + n)$ for coordinate-wise Jacobi. This coordinate-wise methanism also reminds us of Gauss-Seidel method. Recall that Gauss-Seidel:

    **Initialize:** $A = L + U$, where $L$ is lower triangular matrix and $U$ is upper trianglular matrix
    **Iterations:** $\boldsymbol{x}^+ \leftarrow L^{-1}(\boldsymbol{b} - U\boldsymbol{x})$.

---
**Algorithm 4** Coordinate-wise Jacobi Method
---
1: **Input:** Symmetric diagonal dominant matrix $A \in \mathbb{R}^{n \times n}$, vector $\boldsymbol{b} \in \mathbb{R}^n$, number of selected coordinates, $k$, and number of iterations, $L$.
2: Initialize $\boldsymbol{x}^{(0)} \in \mathbb{R}^n$ and set $A = D + R$, where $D$ is diagonal component of $A$ and $R$ is the remainder part. $\boldsymbol{z}^{(0)} = R\boldsymbol{x}^{(0)}$. Set coordinate selecting criterion $\boldsymbol{c}^{(0)} = \boldsymbol{b} - A\boldsymbol{x}^{(0)} = b - D\boldsymbol{x}^{(0)} - \boldsymbol{z}^{(0)}$.
3: **for** $l = 1$ **to** $L$ **do**
4:     Let $\Omega^{(l)}$ be a set containing $k$ coordinates of $\boldsymbol{c}^{(l-1)}$ with the largest magnitude. Execute the following updates:

$$
\begin{aligned}
x_j^{(l)} &= \begin{cases} (b_j - z_j^{(l-1)})/D_{jj}, & j \in \Omega^{(l)} \\ x_j^{(l-1)}, & j \notin \Omega^{(l)} \end{cases} \\
\boldsymbol{z}^{(l)} &= \boldsymbol{z}^{(l-1)} + R(\boldsymbol{x}_{\Omega^{(l)}}^{(l)} - \boldsymbol{x}_{\Omega^{(l)}}^{(l-1)}) \\
\boldsymbol{c}^{(l)} &= \boldsymbol{b} - D\boldsymbol{x}^{(l)} - \boldsymbol{z}^{(l)}
\end{aligned}
$$

5: **Output:** $\boldsymbol{x}^{(L)}$
---

And taking advantage of triangular form, the procedure could be simplified as the following version,

$$
x_i^{(l+1)} \leftarrow \frac{1}{a_{ii}}(b_i - \sum_{j=1}^{i-1} a_{ij}x_j^{(l+1)} - \sum_{j=i+1}^{n} a_{ij}x_j^{(l)})
$$

which is very similar to Jacobi method, but uses a forward substitution on newly computed $x_i$. Therefore our method is also like a greedy block version of Gauss-Seidel method. While Gauss-Seidel is like a cyclic coordinate version of our method.

We use Matlab to do some simple experiments on some synthetic data to measure the convergence time until the error is less than $1e - 5$. Here error is measured by $A$-quadratic norm between current iteration $\boldsymbol{x}^{(l)}$ from ground truth $\boldsymbol{x}^*$, namely, $\|\boldsymbol{x}^{(l)} - \boldsymbol{x}^*\|_A = \sqrt{(\boldsymbol{x}^{(l)} - \boldsymbol{x}^*)^T A(\boldsymbol{x}^{(l)} - \boldsymbol{x}^*)}$.

Table 2: Comparison between Jacobi method and Coordinate-wise Jacobi method. N/A denotes the algorithm doesn't converge.

| Dataset | n | $\frac{\lambda_2}{\lambda_1}(A)$ | $\sigma(D^{-1}R)$ | Flops($/n^2$) | | | Speedup | |
|---|---|---|---|---|---|---|---|---|
| | | | | Jacobi | C-Jacobi | Gauss-Seidel | on Jacobi | on G-S |
| 1 | 1000 | 0.7803 | 0.6870 | 35.035 | **4.794** | 7.007 | **7.308** | **1.462** |
| 2 | 1000 | 0.5565 | 0.9524 | 254.254 | **4.284** | 9.009 | **59.350** | **2.103** |
| 3 | 1000 | 0.5224 | 0.9942 | 2115.113 | **4.488** | 9.009 | **471.282** | **2.007** |
| 4 | 1000 | 0.5206 | 0.9986 | 8505.50 | **4.08** | 9.009 | **2084.68** | **2.2081** |
| 5 | 1000 | 0.495 | **1.11** | $N/A$ | **4.386** | 9.009 | $N/A$ | **2.054** |
| 6 | 5000 | 0.7792 | 0.6948 | 40.01 | **5.321** | 8.002 | **7.519** | **1.504** |
| 7 | 5000 | 0.5443 | 0.9529 | 290.058 | **4.317** | 9.002 | **67.187** | **2.085** |
| 8 | 5000 | 0.5146 | 0.9949 | 2703.54 | **5.622** | 10.002 | **480.852** | **1.779** |
| 9 | 5000 | 0.5111 | 0.9992 | 19760.0 | **6.256** | 10.002 | **3158.76** | **1.599** |
| 10 | 5000 | 0.5063 | **1.02** | $N/A$ | **6.256** | 10.002 | $N/A$ | **1.599** |

And from Table A.8, we can see that coordinate-wise Jacobi method shows significant speedup over the naive Jacobi method. And even when the matrix is no longer diagonal dominant, (see table when $\sigma(D^{-1}R) > 1$), but still positive definite, coordinate-wise Jacobi method still converges. And this trait meets the convergence requirement for Gauss-Seidel method.

Although in this comparison coordinate-wise Jacobi doesn't beat up Gauss-Seidel that much, Gauss-Seidel has the disadvantage that it can not be done in parallel, while our method could be more flexible on that. For example, we could greedily update coordinates in each worker, rather than choosing globally the most greedy coordinates.

However, since for symmetric diagonal dominant matrices, Jacobi or Gauss-Seidel is not the state-of-the-art method for solving linear system, we will need to compare with other more powerful methods. And this algorithm lacks theoretical support at this point, so we consider this as an expansion of our current work on coordinate-wise power method. But still, it's worth mentioning that the coordinate-wise mechanism could be powerful applying to Jacobi method and maybe to other iterative methods in linear algebra too. Therefore in the future, we may continue exploiting the theory behind, and analyze why and how greediness impacts on Jacobi method or other iterative methods in linear algebra.