[Reviews · NeurIPS 2016]

Reviewer 1

Summary

The paper propose a coordinate wise power method computing the dominant eigenvector of a matrix.

Qualitative Assessment

The paper is nicely written. It has novel results and should be of interest to many in the community. I have only a minor comment. How is the behavior of CPM (Algorithm 1) for a symmetric negative definite matrix A.

Confidence in this Review

2-Confident (read it all; understood it all reasonably well)


Reviewer 2

Summary

The paper presents two coordinate-wise power methods to compute the largest eigenvector of a matrix. The motivation is based on the observation that, in normal power method, some coordinates will converge quite fast to the final result while some will be slow. So a natural question is whether one can select the coordinates with potential large changes in the next iteration and with small overhead. This paper gives an affirmative answer via two coordinate methods. Theoretical analysis on the convergence behavior is also provided, which indicates the proposed methods have the similar iteration complexity as the basic power method, but its cost per iteration is smaller. Empirical studies on several real graphs, from small to large, have shown the proposed method has significant efficiency improvement over the vanilla power method.

Qualitative Assessment

This is an interesting paper addressing a fundamental problem in eigenvector computation. Normally, people aim to accelerate the power method from a lot of aspects, but seem rarely pay attention to the phenomenon that different coordinates can have various convergence rate. This paper appears to be the first to solve this problem, and make a connection to coordinate decent. The criterion to select the next set of coordinates are intuitive, i.e., picking those maximizing the difference between the current iterate and the next. Both theoretical and empirical results look reasonable as well. I think the point of view to look at the problem sounds novel and should be appreciated. As power method is the building block of quite a few applications in machine learning, linear algebra and graph computation, this work may potentially have large impact.

Confidence in this Review

2-Confident (read it all; understood it all reasonably well)


Reviewer 3

Summary

A coordinate descent method of the power method is proposed with convergence analysis. An even faster algorithm based on greedy coordinate descent is proposed. Empirical results suggest that the proposed method is fast in practice compared to the baseline or vanilla power method. The authors show that the algorithms proposed have a local linear convergence.

Qualitative Assessment

Overall I enjoyed reading the paper, it is well written and flows well with the intuition for the algorithms clearly written in the main paper which is helpful. However I am not sure how the comparison is fair: the convergence analysis only applies to the problem of computing eigenvectors. If this is the case, it is important to compare with recent algorithms that compute the same, comparing just with the vanilla power method does not say much about usefulness of the proposed algorithms, many state-of-the-art algorithms are often randomized/parallelized see for example [1], [2], [3] (I am not familiar with recent papers in SIAM, Matrix analysis, Numerical Linear Algebra but these seem like good baselines to compare with). A discussion of how the proposed algorithms can or cannot be parallelized WITH convergence would greatly enhance the utility. The authors claim that their method is useful when the entire matrix cannot fit in the memory but is not made concrete: usually in this scenario, communication costs become significant see [4]. Finally, local convergence is already known, see [5] although showing that the function is strongly convex is still a valid contribution. 1. A Parallel Eigensolver for Dense Symmetric Matrices based on Multiple Relatively Robust Representations (2005) by P. Bientinesi, I. S. Dhillon, and R. A. van de Geijn; 2. Finding Structure With Randomness: Probabilistic Algorithms For Constructing Approximate Matrix Decompositions (2010) by N. Halko, P. G. Martinsson AND J. A. Tropp. 3. An Algorithm for the Principal Component Analysis of large Data Sets (2011) by N. Halko, P.G. Martinsson, Y. Shkolnisky, M. Tygert. 4. Communication-Avoiding Krylov Subspace Methods in Theory and Practice (2015) PhD Thesis by Erin Carson, URL = {http://www.eecs.berkeley.edu/Pubs/TechRpts/2015/EECS-2015-179.html} 5. Coordinate Descent Converges Faster with the Gauss-Southwell Rule Than Random Selection (ICML 2015) by Julie Nutini, Mark Schmidt, Issam Laradji, Michael Friedlander, Hoyt Koepke

Confidence in this Review

2-Confident (read it all; understood it all reasonably well)


Reviewer 4

Summary

The paper proposes a coordinate version of the power method. A selection rule is provided to select the most relevant coordinates to update. Some applications are discussed and results are shown to outperform the simple power method. A thorough analysis is of the convergence behavior is also provided.

Qualitative Assessment

The paper considers an important improvement of the power method, and a coordinate version is proposed. My worries are related to the not-so-strong improvements on the power method in terms of CPU time (Figure 2 in particular), the insensitivity to parameter k (which indicates that a simple random coordinate descent would be as good as the proposed method). I am also worried about the time improvements shown in Table 1. While Figure 2 does not show much improvement in terms of CPU time, i wonder how such drastic improvements are possible in Table 1? Are these improvements artifacts of the objective function in eqn. (4) or are they due to the coordinate power method? A clear discussion about this is required. The application considered in the paper is minimal and uninteresting. Since the convergence analysis somehow depends on the results from ref. [6], I would urge the authors to strongly consider applications mentioned in [6] and their variants. This would bring much more value to the paper. Other applications which involve computing an inverse of a large sparse/dense matrix could also be tried. Why not extend the general analysis provided in ref. [6] to the proposed method? That is, the authors can assume that a noise factor is added to the coordinates in the set \Omega and provide a unified analysis. I strongly urge the authors to improve the paper along these directions. *************Update after reading authors' rebuttal ***************** I have read the authors' rebuttal. I stand by my review and do not wish to change my scores.

Confidence in this Review

2-Confident (read it all; understood it all reasonably well)


Reviewer 5

Summary

The paper proposes two algorithms of coordinate descent power methods for finding the maximum eigenvectors. Inspired by the observations that different coordinates converge at different speeds, the coordinate descent power method chooses the most prominent coordinates greedily to update. The paper provides convergence analysis for the proposed algorithms and demonstrated the speed advantage on a variety of datasets.

Qualitative Assessment

The algorithms seem practical and achieve some speedup compared with vanilla power methods. The experiments are comprehensive in terms of the variety of datasets, and the out-of-core part seems interesting. In terms of novelty, there are some contributions in terms of the insight that different coordinates converge at different rates and coordinate-descent methods should run faster. However, the novelty is limited in the sense it is application of coordinate descent on power iterations. The proof techniques seem standard, based on noisy power methods. The experiments look promising based on current comparisons since the vanilla power method is simple. It would be more interesting if the authors can compare with other large scale eigenvalue solvers such as randomized SVD. Some comment on Theorem 4.1. It is difficult to see the convergence rate in its current form. It would be more enlightening to phrase it as the number of iterations required to achieve a certain error. Or the authors can provide more explanations on how to interpret result. The quantity phi^(l)(k) and 1/cos(theta) makes it hard to see the exact convergence rate. One question on Algorithm 1. In the step x^(l) = y^(l) / || y^(l), all the elements of x^(l) are changed due to normalization, then in Eqn (3), the difference in x^(l) and x^(l-1) restricted on the set Omega^(l) is no longer equal to the difference without such restriction. How can you still maintain z = A x in this case?

Confidence in this Review

3-Expert (read the paper in detail, know the area, quite certain of my opinion)


Reviewer 6

Summary

The paper combines (block) coordinate descent-techniques with power methods to propose an efficient method computing dominant eigenvector. The proposed method is simple but more efficient than simple power methods. The authors also give detailed theoretical analyses evaluating which guarantees the performance of the algorithm. The efficiency of the proposed method are confirmed also by numerical experiments.

Qualitative Assessment

The paper is very pleasant to read regarding its clear explanation and logic flow. The proposed algorithm is quite simple, but is proved to much more efficient than the power method under some conditions. Although the authors claims that the performance is not sensitive to the choice of k, I believe that it is possible to tune k for the improved performance, especially when |\lambda_1/\lambda_2| is close to 0 or 1.

Confidence in this Review

1-Less confident (might not have understood significant parts)